# Prediction of the Epidemic Peak of Coronavirus Disease in Japan, 2020

**DOI:** 10.3390/jcm9030789

**Published:** 2020-03-13

**Authors:** Toshikazu Kuniya

**Affiliations:** Graduate School of System Informatics, Kobe University, 1-1 Rokkodai-cho, Nada-ku, Kobe 657-8501, Japan; tkuniya@port.kobe-u.ac.jp

**Keywords:** COVID-19, SEIR compartmental model, basic reproduction number

## Abstract

The first case of coronavirus disease 2019 (COVID-19) in Japan was reported on 15 January 2020 and the number of reported cases has increased day by day. The purpose of this study is to give a prediction of the epidemic peak for COVID-19 in Japan by using the real-time data from 15 January to 29 February 2020. Taking into account the uncertainty due to the incomplete identification of infective population, we apply the well-known SEIR compartmental model for the prediction. By using a least-square-based method with Poisson noise, we estimate that the basic reproduction number for the epidemic in Japan is $R_{0}=2.6$ (95%CI, 2.4–2.8) and the epidemic peak could possibly reach the early-middle summer. In addition, we obtain the following epidemiological insights: (1) the essential epidemic size is less likely to be affected by the rate of identification of the actual infective population; (2) the intervention has a positive effect on the delay of the epidemic peak; (3) intervention over a relatively long period is needed to effectively reduce the final epidemic size.

## 1. Introduction

In December 2019, the first case of respiratory disease caused by a novel coronavirus was identified in Wuhan City, Hubei Province, China. The outbreak of the disease is ongoing worldwide and the World Health Organization named it coronavirus disease 2019 (COVID-19) on 11 February 2020 [1]. In Japan, the first case was reported on 15 January 2020 and the number of reported laboratory-confirmed COVID-19 cases per week has increased day by day (see Table 1).

As seen in Table 1, the number of newly reported cases per week has increased and a serious outbreak in Japan is a realistic outcome. One of the greatest public concerns is whether the epidemic continues until summer so that it affects the Summer Olympics, which is planned to be held in Tokyo. The purpose of this study is to give a prediction of the epidemic peak of COVID-19 in Japan, which might help us to act appropriately to reduce the epidemic risk.

The epidemic data as shown in Table 1 would have mainly twofold uncertainty. The first one is due to the fact that asymptomatic infected people could spread the infection [3]. The second one is due to the lack of opportunity for the diagnostic test as sufficiently simple diagnostic test kits have not been developed yet and the diagnosis in the early stage in Japan was mainly restricted to people who visited Wuhan [4]. In this study, taking into account such uncertainty, we apply a simple and well-known mathematical model for the prediction. More precisely, we assume that only *p* (0<p≤1) fraction of infective individuals can be identified by diagnosis.

## 2. Methods

### 2.1. Model

We apply the following well-known SEIR compartmental model (see, e.g., [5]) for the prediction.
(1)$$\left\{\begin{array}{c} S^{\prime }\left(t\right)=-\beta S\left(t\right)I\left(t\right),\;E^{\prime }\left(t\right)=\beta S\left(t\right)I\left(t\right)-\varepsilon E\left(t\right),\\ I^{\prime }\left(t\right)=\varepsilon E\left(t\right)-\gamma I\left(t\right),\;R^{\prime }\left(t\right)=\gamma I\left(t\right),\;t>0, \end{array}\right.$$
where S(t), E(t), I(t) and R(t) denote the susceptible, exposed, infective and removed populations at time *t*, respectively. β, ε and γ denote the infection rate, the onset rate and the removal rate, respectively. Note that 1/ε and 1/γ imply the average incubation period and the average infectious period, respectively. Let the unit time be 1 day. Based on the previous studies [6,7], we fix 1/ε=5, and thus, ε=0.2 and γ=0.1, respectively. We fix S+E+I+R to be 1 so that each population implies the proportion to the total population. We assume that one infective person is identified at time t=0 among total $N=1.26\times 10^{8}$ number of people in Japan [8]. That is, $Y\left(0\right)=pI\left(0\right)\times 1.26\times 10^{8}=1$, where
$$Y\left(t\right)=pI\left(t\right)\times 1.26\times 10^{8}$$
denotes the number of infective individuals who are identified at time *t*. Thus, we obtain $I\left(0\right)=1/\left(p\times 1.26\times 10^{8}\right)$. We assume that there is no exposed and removed populations at t=0, that is, E(0)=R(0)=0, and hence,
$$S\left(0\right)=1-E\left(0\right)-I\left(0\right)-R\left(0\right)=1-\frac{1}{p\times 1.26\times 10^{8}}.$$

It was estimated in [9] that 77 cases were confirmed among the possible 940 infected population in February in Hokkaido, Japan. Based on this report, we assume that *p* ranges from 0.01 to 0.1. The basic reproduction number $R_{0}$, which means the expected value of secondary cases produced by one infective individual [10], is calculated as the maximum eigenvalue of the next generation matrix $FV^{-1}$ [11], where
$$F=\left[\begin{array}{cc} 0 & \beta S(0)\\ 0 & 0 \end{array}\right],\;V=\left[\begin{array}{cc} \varepsilon & 0\\ -\varepsilon & \gamma \end{array}\right].$$

Thus, we obtain
(2)$$R_{0}=\frac{\beta S(0)}{\gamma }=\frac{\beta }{\gamma }\left(1-\frac{1}{p\times 1.26\times 10^{8}}\right).$$

### 2.2. Sensitivity of the Basic Reproduction Number

It is obvious that the basic reproduction number $R_{0}$ is independent from the onset rate ε. The sensitivity of $R_{0}$ to other parameters β, γ and *p* are calculated as follows:(3)$$A_{\beta }=\frac{\beta }{R_{0}}\frac{\partial R_{0}}{\partial \beta }=1,\;A_{\gamma }=\frac{\gamma }{R_{0}}\frac{\partial R_{0}}{\partial \gamma }=-1,\;A_{p}=\frac{p}{R_{0}}\frac{\partial R_{0}}{\partial p}=\frac{1}{p\times 1.26\times 10^{8}-1},$$
where $A_{\beta }$, $A_{\gamma }$ and $A_{p}$ denote the normalized sensitivity indexes with respect to β, γ and *p*, respectively. We see from Equation (Equation 3) that the *k* time’s increase in β (resp. γ) results in the *k* (resp. $k^{-1}$) time’s increase in $R_{0}$. In particular, we see from the third equation in Equation (Equation 3) that $A_{p}\approx 0$ if $p\geq 1.0\times 10^{-6}$. This implies that the identification rate *p* in a realistic range almost does not affect the size of $R_{0}$.

### 2.3. Estimation of the Infection Rate

Let y(t), t=0,1,…,45 be the number of daily reported cases of COVID-19 in Japan from 15 January (t=0) to 29 February (t=45) 2020. We perform the following least-square-based procedure with Poisson noise to estimate the infection rate β.

**Description** **1****.** *(P1) Fix*β>0*and calculate the numerical value of *Y(t), t=0,1,…,45*by using model Equation* (Equation 1).*(P2) Calculate*$$\tilde{Y}\left(t\right)=Y\left(t\right)+\sqrt{Y(t)}\epsilon \left(t\right)=Y\left(t\right)+\left(\mathrm{Poisson}\;\mathrm{noise}\right),\;t=0,1,\ldots ,45,$$*where *ϵ(t), t=0,1,…,45*denote random variables from a normal distribution with mean zero and variance* 1 *[12]*.*(P3) Calculate*$J\left(\beta \right)=\sum_{t=0}^{45}{\left[y\left(t\right)-\tilde{Y}\left(t\right)\right]}^{2}$.*(P4) Run (P1)–(P3) for*0.2≤β≤0.4*and find*$\beta^{*}$*such that*$J\left(\beta^{*}\right)=\min_{0.2\leq \beta \leq 0.4}J\left(\beta \right)$.*(P5) Repeat (P1)–(P4)*10,000*times and obtain the distribution of*$\beta^{*}$.*(P6) Approximate the distribution of*$\beta^{*}$*by a normal distribution and obtain a*95%*confidence interval*.

Note that for the reason stated above, the value of 0.01≤p≤0.1 does not affect this estimation procedure. By (P1)–(P6), we obtain a normal distribution with mean 0.26 and standard derivation 0.01. Thus, we obtain an estimation of β as 0.26 (95%CI, 0.24–0.28) (see Figure 1). Moreover, by Equation (Equation 2), we obtain an estimation of $R_{0}$ as 2.6 (95%CI, 2.4–2.8) (see Table 2).

## 3. Results

### 3.1. Peak Prediction

We define the epidemic peak $t^{*}$ by the time such that *Y* attains its maximum in 1 year, that is, $Y\left(t^{*}\right)=\max_{0\leq t\leq 365}Y\left(t\right)$. We first set p=0.1. In this case, we obtain the following figure on the long time behavior of Y(t) for β=0.28, 0.26 and 0.24.

We see from Figure 2 that the estimated epidemic peak is $t^{*}=208$ (95%CI, 191–229). That is, starting from 15 January (t=0), the estimated epidemic peak is 10 August (t=208) and the uncertainty range is from 24 July (t=191) to 31 August (t=229).

We next set p=0.01. In this case, we obtain the following figure.

We see from Figure 3 that the estimated epidemic peak is $t^{*}=179$ (95%CI, 165–197). That is, starting from January 15 (t=0), the estimated epidemic peak is July 12 (t=179) and the uncertainty range is from June 28 (t=165) to July 30 (t=197). In contrast to $R_{0}$, the epidemic peak and the (apparent) epidemic size are sensitive to the identification rate *p*. Note that the essential epidemic size, which is characterized by $R_{0}$, is almost the same in both of p=0.1 and p=0.01.

### 3.2. Possible Effect of Intervention

We next discuss the effect of intervention. In Japan, school closure has started in almost all prefectures from the beginning of March [13] and many social events have been cancelled off to reduce the contact risk. However, the exact effect of such social efforts is unclear and might be limited as the proportion of young people to the whole infected people of COVID-19 seems not so high (2% of 72,314 reported cases in China [14]). In this simulation, we assume that such social efforts successfully reduce the infection rate β=0.26 to 75% during a period from 1 March (t=46) to a planned day (t=T≥47). In what follows, we fix p=0.01.

First, we set T=77, that is, the intervention is carried out for 1 month (from 1 March to 1 April). In this case, the epidemic peak $t^{*}$ is delayed from 179 (12 July) to 190 (23 July). However, the epidemic size is almost the same. On the other hand, if T=220, that is, the intervention is carried out for 6 months (from 1 March to 1 September), then the epidemic peak $t^{*}$ is delayed from 179 (12 July) to 243 (14 September) and the epidemic size is effectively reduced (see Figure 4).

More precisely, we see from Figure 5a that the epidemic peak $t^{*}$ is delayed almost linearly for 47≤T≤239 and fixed to $t^{*}=237$ for T≥240.

This implies that the intervention has a positive effect on the delay of the epidemic peak, which would contribute to improve the medical environment utilizing the extra time period. On the other hand, we see from Figure 5b that the number of accumulated cases at t=365, which is calculated as $pR\left(365\right)\times 1.26\times 10^{8}$, is monotonically decreasing and converges to $0.99\times 10^{6}$ as *T* increases. However, it almost does not change for small T≤180. This implies that the intervention over a relatively long duration is required to effectively reduce the final epidemic size.

## 4. Discussion

In this study, by applying the SEIR compartmental model to the daily reported cases of COVID-19 in Japan from 15 January to 29 February, we have estimated that the basic reproduction number $R_{0}$ is 2.6 (95%CI, 2.4–2.8) and the epidemic peak could possibly reach the early-middle summer. Of course, this kind of long range peak prediction would contain the essential uncertainty due to the possibility of some big changes in the social and natural (climate) situations. Nevertheless, our result suggests that the epidemic of COVID-19 in Japan would not end so quickly. This might be consistent with the WHO’s statement on 6 March 2020 that it is a false hope that COVID-19 will disappear in the summer like the flu [15].

The estimated value of the basic reproduction number $R_{0}$ in this study is not so different from early estimations: 2.6 (95%CI, 1.5–3.5) [16], 2.90 (95%CI, 2.32–3.63) [17], 3.11 (95%CI, 2.39–4.13) [18], 3.58 (95%CI, 2.89–4.39) [19] and 3.28 (average of estimations in 12 studies) [20]. In addition, in this study, we have obtained the following epidemiological insights:The essential epidemic size, which is characterized by $R_{0}$, would not be affected by the identification rate *p* in a realistic parameter range 0.01–0.1, in particular, $p\geq 1.0\times 10^{-6}$.The intervention exactly has a positive effect on the delay of the epidemic peak, which would contribute to improve the medical environment utilizing the extra time period.Intervention over a relatively long period is needed to effectively reduce the final epidemic size.

The first statement implies that underestimation of the actual infective population would not contribute to the reduction of the essential epidemic risk. Correct information based on an adequate diagnosis system would be desired for people to act appropriately.

## Figures and Tables

**Figure 1 jcm-09-00789-f001:**
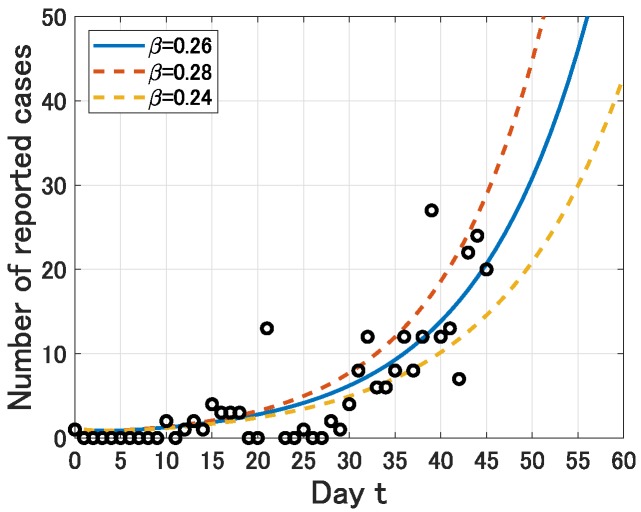
Comparison of Y(t) with the estimated infection rate β and the number of daily reported cases of COVID-19 in Japan from 15 January (t=0) to 29 February (t=45).

**Figure 2 jcm-09-00789-f002:**
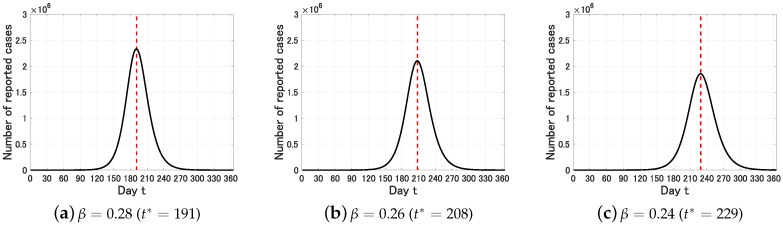
Time variation of the number Y(t) of infective individuals who are identified at time *t* (0≤t≤365) for p=0.1. The dot lines represent the epidemic peak $t^{*}$.

**Figure 3 jcm-09-00789-f003:**
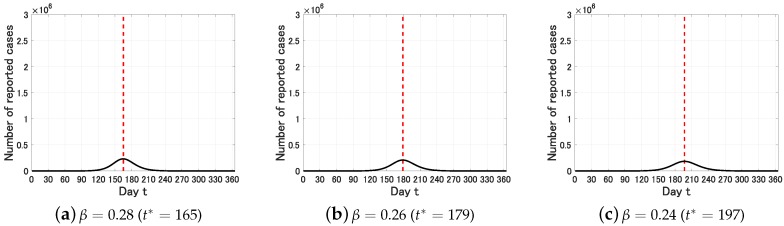
Time variation of the number Y(t) of infective individuals who are identified at time *t* (0≤t≤365) for p=0.01. The dot lines represent the epidemic peak $t^{*}$.

**Figure 4 jcm-09-00789-f004:**
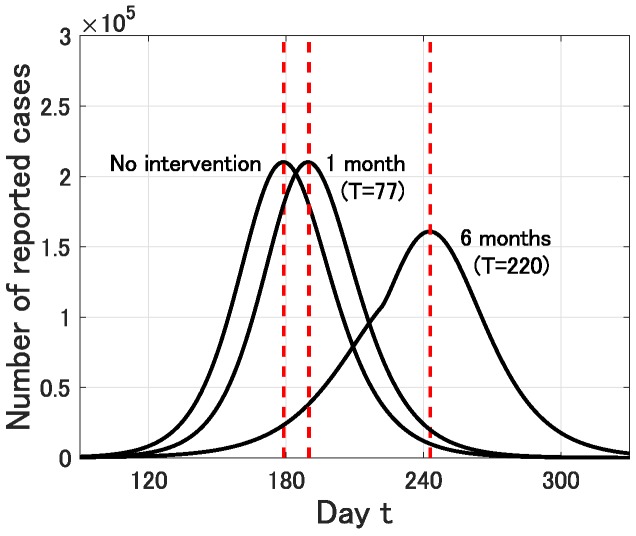
Time variation of the number Y(t) of infective individuals who are identified at time *t* (0≤t≤365) for p=0.01 and no intervention, 1 month intervention (T=77) and 6 months intervention (T=220). The dot lines represent the epidemic peak.

**Figure 5 jcm-09-00789-f005:**
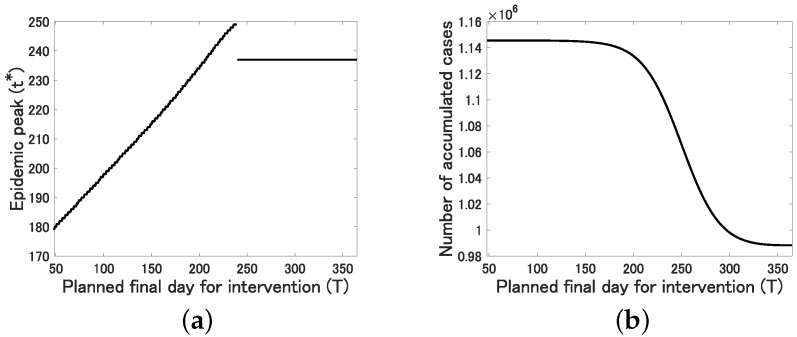
The relation between the planned final day for intervention *T* and (**a**) the epidemic peak $t^{*}$; (**b**) the number of accumulated cases at time t=365: $pR\left(365\right)\times 1.26\times 10^{8}$.

**Table 1 jcm-09-00789-t001:** Number of newly reported COVID-19 cases in Japan until 1 March 2020 [2].

Week	Number of Newly Reported Cases	Number of Accumulated Cases
12 January–18 January	1	1
19 January–25 January	2	3
26 January–1 February	14	17
2 February–8 February	8	25
9 February–16 February	28	53
17 February–23 February	79	132
24 February–1 March	107	239

**Table 2 jcm-09-00789-t002:** Parameter values for model Equation (Equation 1).

Parameter	Description	Value	Reference
β	Infection rate	0.26 (95%CI, 0.24–0.28)	Estimated
$R_{0}$	Basic reproduction number	2.6 (95%CI, 2.4–2.8)	Estimated
ε	Onset rate	0.2	[6]
γ	Removal rate	0.1	[7]
*N*	Total population in Japan	$1.26\times 10^{8}$	[8]
*p*	Identification rate	0.01–0.1	[9]

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
