# Peer review of "Prediction of the Epidemic Peak of Coronavirus Disease in Japan, 2020"

_jcm, 2020, doi:10.3390/jcm9030789_

Round 1

Reviewer 1 Report

This work uses identified infections of covid19 by reporting date in Japan, to fit an SIR model, from which a longer-term prediction is made. The topic is certainly of great interest, but I feel there are a number of improvements that need to be made to the work here.

  1. The outbreak of the virus on the Diamond Princess cruise liner has its own particular setting and transmission dynamics, and this is unlikely to reflect transmission 'on the ground' in Japan, so combing the two sources of diagnosed cases will lead to an R0 value which applies to neither scenario. It would be better to identify R0 for each setting independently. 
  2. The model really should be SEIR to account for timing in which individuals are incubation or the infection is latent, before they become symptomatic and may be diagnoses and counted. There is description of the incubation period distribution from a number of sources now. I suggest the variation in the distribution is also accounted for in the analysis, at least as a sensitivity. 
  3. No details are described as to how the parameter values are inferred and the model fitting process
  4. Finally, the parameter estimation and most importantly the prediction need to account for uncertainty both in parameter estimation, but also the potential noise that could be generated through stochastic transmission; this latter element could be incorporated through the addition of Poisson noise at each time step. At least one early covid19 modelling study has done something similar.

The model could be improved further by using daily case data, rather than weekly, if available. Similarly rather than relying on reporting dates, using symptom onset dates would be preferable, again if available. If they are not, they could be imputed, again by reference to other studies (with appropriate caveats). 

Also, there are potentially many large changes in transmission that may occur between the time period covered here and several month hence, not least large scale control or spontaneous public responses such as social distancing. This should be strongly emphasised in the discussion and conclusions regarding any prediction.

Reviewer 2 Report

Very interesting article. The SIR model is a relevant tool for predicting the epidemic curve and size if the force of infection (basic reproduction number) and average duration of infection are known. Although your report uses the model in an appropriate way, there is one thing we should not miss in predicting the size and timing of the epidemic in the real world. Japanese government has already implemented an action to contain the outbreak, such as airplane restriction to China. News media also continuously broadcast the severity and clinical characteristics of the virus on TV and internet. Making a poster also changes people's behavior against the viral infection, leading to smaller possibility of secondary infections in the community. But the parameter you use in your analysis is fixed through time. China, for example, implemented transport restriction within the country and the incidence starts to decrease (please see link below). Other countries like Singapore also started monitoring the confirmed cases and they performed a thorough contact tracing.

https://www.ecdc.europa.eu/en/geographical-distribution-2019-ncov-cases

I would like to ask you to do the analysis considering such intervention effect. It would be interesting if you compared what we could have observed with and without intervention.

Round 2

Reviewer 1 Report

Thank you for addressing my concerns. The manuscript is much improved.

Author Response

I deeply appreciate your review on the manuscript.

Reviewer 2 Report

Muchl improved with the intervention effect taken into account. I would recommend you to state how you came up with the reduction rate of 75% in the infection rate.

Author Response

I deeply appreciate your review on the manuscript. I added the following statement to line 70,

"However, the effect of such social efforts is unclear and might be limited as the proportion of young people to the whole infected people of COVID-19 seems not so high (2% of 72,314 reported cases in China [14])."

citing the following reference.

[14] Wu, Z.; McGoogan, J.M. Characteristics of and imoprtant lessons from the coronavirus disease 2019 (COVID-19) outbreak in China. J. Amer. Med. Assoc. 2020. DOI: 10.1001/jama.2020.2648